# Impact of the COVID-19 Pandemic Surveillance of Visceral Leishmaniasis in Brazil: An Ecological Study

**Josefa Rayane Santos Silveira** [1], **Shirley Verônica Melo Almeida Lima** [1,2], **Allan Dantas dos Santos** [1,2], **Luana Silva Siqueira** [2], **Guilherme Reis de Santana Santos** [2], **Álvaro Francisco Lopes de Sousa** [3,*], **Layze Braz de Oliveira** [4], **Isabel Amélia Costa Mendes** [4] **and Caíque Jordan Nunes Ribeiro** [1,2]

1 Graduate Program in Nursing, Federal University of Sergipe, São Cristóvão 49100-000, SE, Brazil; rayanesilveira@academico.ufs.br (J.R.S.S.); shirleylima@academico.ufs.br (S.V.M.A.L.); allanufs@academico.ufs.br (A.D.d.S.); caiquejordan@academico.ufs.br (C.J.N.R.)
2 Nursing Department, Federal University of Sergipe, Lagarto 49400-000, SE, Brazil; luanasilva21@academico.ufs.br (L.S.S.); guilheermereeis@gmail.com (G.R.d.S.S.)
3 Hospital Sírio-Libânes, Instituto de Ensino e Pesquisa, São Paulo 01308-050, SP, Brazil
4 Nursing School of Ribeirão Preto, Centro Colaborador da OPAS/OMS para o Desenvolvimento da Pesquisa em Enfermagem, Universidade de São Paulo, Ribeirão Preto 14040-902, SP, Brazil; layzebraz@gmail.com (L.B.d.O.); iamendes@usp.br (I.A.C.M.)
* Correspondence: sousa.alvaromd@gmail.com

**Abstract:** The aim of the study was to assess the impact of the COVID-19 pandemic on the notification of new VL cases in Brazil in 2020. It is an ecological and time-series study (2015–2020) with spatial analysis techniques, whose units of analysis were the 5570 Brazilian municipalities. The study population consisted of all new cases of VL recorded between 2015 and 2020. The P-score was calculated to estimate the percentage variation in new VL cases. Global and local univariate Moran's Indices and retrospective space–time scan statistics were used in spatial and space–time analyses, respectively. It was expected that there would be 3627 new cases of VL in Brazil in 2020, but 1932 cases were reported (−46.73%). All Brazilian regions presented a negative percentage variation in the registration of new VL cases, with the Southeast (−54.70%), North (−49.97%), and Northeast (−44.22%) standing out. There was spatial dependence of the disease nationwide in both periods, before and during the first year of the COVID-19 pandemic. There was a significant reduction in the incidence of new VL cases in Brazil during the first year of the COVID-19 pandemic. These findings reinforce the need for better preparedness of the health system, especially in situations of new epidemics.

**Keywords:** spatial analysis; spatiotemporal analysis; COVID-19; time series study; visceral leishmaniasis

## 1. Introduction

COVID-19 rapidly spread worldwide and was declared a pandemic in early 2020. In Brazil, due to its vast geographical expanse and significant economic, social, and cultural disparities, the impact of the disease was unevenly manifested across the country [1–3].

The burden placed on the Brazilian healthcare system to combat the pandemic, coupled with the geographical overlap between COVID-19 and other diseases, may have weakened strategies for the surveillance and control of neglected tropical diseases (NTDs), such as visceral leishmaniasis (VL) [4].

VL is a vector-borne parasitic disease that continues to expand persistently within the country [5,6]. Socio-environmental changes, such as the impoverishment of the population, lack of urban planning, migration, climate change, and precarious living conditions, along with inadequate basic sanitation, contribute to shifts in epidemiological patterns and the ongoing spread of VL in the region [7–9]. These factors result in increased vulnerability among the population, which can be further exacerbated by COVID-19 [10].

In this context, it is believed that the COVID-19 pandemic has had far-reaching impacts on health systems and programs, particularly among populations where NTDs are endemic [11], especially with regard to the registration of new cases and early treatment.

Although it is a fundamental component of epidemiological surveillance, the notification and confirmation process for cases may be sluggish [5], leading to poorer outcomes for VL patients, as the disease is fatal in over 95% of cases when not diagnosed and treated promptly [9].

Spatial–temporal analysis techniques can be valuable tools in this context, particularly for monitoring and prioritizing high-risk areas. Disruptions in VL surveillance efforts due to the pandemic may have affected case registration. In this regard, the identification of public health issues may have been delayed, posing a challenge to the healthcare system's response [11]. Therefore, this study aims to assess the impact of the COVID-19 pandemic on the reporting of new cases of VL in Brazil in 2020.

## 2. Materials and Methods

### 2.1. Study Design

This is an ecological and time-series study (2015–2020) that used spatial, temporal, and space–time analysis techniques, whose units of analysis were the 5570 Brazilian municipalities.

### 2.2. Study Area

Brazil is the largest country in South America and divided into five geographic regions (North, Northeast, Central-west, Southeast, and South). It has 27 federative units, including one federal district, 26 states, and 5570 municipalities (Figure 1) [12].

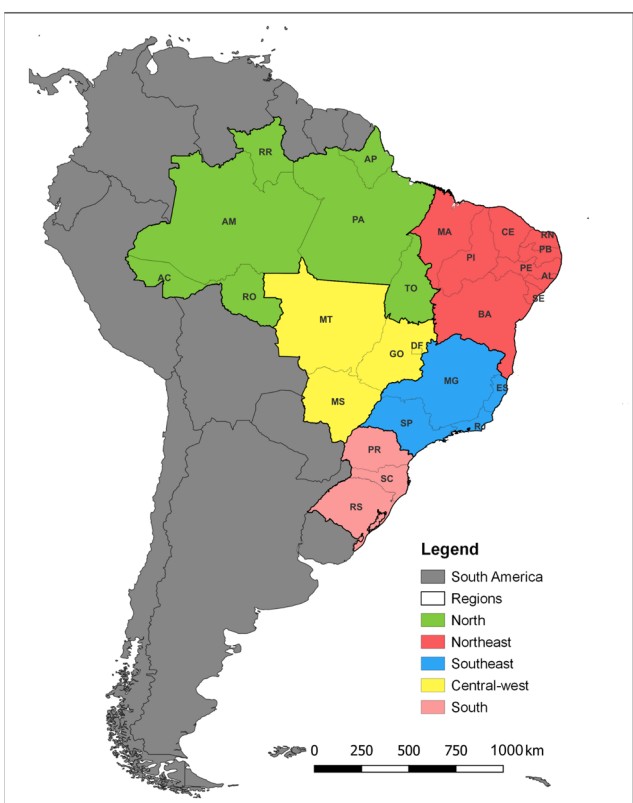

**Figure 1.** Characterization of the study area. AC: Acre; AL: Alagoas; AM: Amazonas; AP: Amapá; BA: Bahia; CE: Ceará; DF: Distrito Federal; ES: Espírito Santo; GO: Goiás; MA: Maranhão; MG: Minas Gerais; MT: Mato Grosso; MS: Mato Grosso do Sul; PA: Pará; PB: Paraíba; PE: Pernambuco; PI: Piauí; PR: Paraná; RJ: Rio de Janeiro; RN: Rio Grande do Norte; RO: Rondônia; RR: Roraima; RS: Rio Grande do Sul; SC: Santa Catarina; SE: Sergipe; SP: São Paulo; TO: Tocantins.

### 2.3. Study Population and Data Source

The study population consisted of all new cases of VL reported in Brazil between 2015 and 2020. Records without municipal allocation were excluded. Data on new cases of VL were obtained from the Notifiable Diseases Information System (SINAN) of the Department of Informatics of the Unified Health System (DATASUS). Population data for municipalities were extracted from the Brazilian Institute of Geography and Statistics (IBGE) database, considering information from the national population census of 2010 and official estimates for the intercensal years. The digital cartographic mesh of Brazil was obtained from the IBGE in shapefile format, in the Geographic Projection System in latitude/longitude.

### 2.4. Study Variables

The P-score was considered the primary variable in this study for making inferences about the excess or deficit of new VL case notifications. By comparing the expected and observed values, it is possible to calculate the increase or decrease in the occurrence of the phenomenon over time and space [13].

$$\text{P-Score} = \frac{\text{Number of cases reported in 2020} - \text{expected number of new cases in 2020}}{\text{Expected number of new cases in 2020}} \times 100$$

The number of expected cases for 2020 corresponds to the average of cases reported in the five years prior to the year under analysis (2015–2019). As a result, positive percentage values indicate an increase in the number of cases, and negative values indicate a decrease in the number of cases compared to expected values [13]. The annual P-score was calculated at the municipal, state, regional, and national levels. The monthly P-score was calculated at the state level. Additionally, we also calculated the P-scores at the state level from 2015 to 2022 in order to demonstrate the decreasing trend of new VL case notifications alongside the pandemic process (Supplementary Data).

### 2.5. Descriptive Analysis

Initially, a descriptive analysis was performed to characterize the clinical-epidemiological profile of new cases of VL. The following variables were described: gender, age group, skin color, region of residence, education level, percentage of cases of VL-HIV coinfection, and outcomes. The data were presented in simple and relative frequencies in tables. P-scores were calculated at the municipal, state, regional, and national levels and represented in graphs and maps.

### 2.6. Spatial Analysis

A descriptive spatial analysis of the spatial data was performed by mapping the spatial distribution of the P-score, the average incidence rates of VL from 2015 to 2019, and the corresponding rates for the year 2020. The stratification of the P-score was performed using equal intervals, and the incidence rates were standardized according to the classification of the Epidemiological Surveillance Guide of the Brazilian Ministry of Health: sporadic transmission (<2.4 cases/100,000 inhabitants), moderate transmission ($\geq$2.4 and <4.4 cases/100,000 inhabitants), and intense transmission ($\geq$4.4 cases/100,000 inhabitants) [1]. Additional mapping is available in Supplementary Data to depict the annual distribution of the P-score at the state level and the VL incidence rate from 2015 to 2022.

To identify the existence of spatial dependence, the univariate Global Moran Index (I) was used, considering a proximity matrix developed with the criterion of first-order contiguity. This index varies from $-1$ to $+1$, where values close to zero indicate spatial randomness; positive values indicate positive spatial autocorrelation; and negative values indicate negative autocorrelation [14].

With the identification of significant spatial autocorrelation, the univariate Local Moral Index (Local Indicators of Spatial Association—LISA) analysis was performed. This method allows for the identification of the occurrence of spatial clusters of municipalities, by generating a scattering diagram with four quadrants (Q1: high/high or hotspots;

Q2: high/low; Q3: low/high; and Q4: low/high). Moran maps were used to represent the municipalities with statistically significant local indicators ($p < 0.05$) [14].

*2.7. Spatiotemporal Analysis*

The identification of high- and low-risk space–time clusters of new VL cases throughout the months of 2020 was carried out through retrospective scanning statistics using the Poisson probability distribution model, using the following parameters: one-month aggregation time; no overlap of geographic or temporal clusters; circular clusters; a maximum spatial cluster size of 50% of the at-risk population; and a maximum temporal cluster size equal to 50% of the study period.

The primary and secondary clusters were detected through the likelihood ratio test (LLR) and represented in the form of maps and tables. The relative risks (RRs) of VL occurrence for each cluster were calculated in relation to their neighbors. Results were considered significant when the *p*-value was <0.05, using 999 Monte Carlo simulations [15].

## 3. Results

Despite an expected number of 3627 new cases of VL in Brazil in 2020, only 1932 were reported, representing a reduction of 46.73% (Figure 2). Similarly, all Brazilian regions showed a negative percentage variation in the registration of new cases of VL, with the Southeast (−54.70%), North (−49.97%), and Northeast (−44.22%) standing out. Most Brazilian states reported a decrease in the notification of new VL cases in 2020 (*n* = 21; 77.8%), with the most significant decreases observed in Rondônia and Santa Catarina (−100%), Paraná (−73.68%), and Minas Gerais (−58.44%). On the other hand, four states reported an increase in new cases, namely Amapá (150%), Santa Catarina (66.67%), Rio Grande do Sul (45.83%), and Alagoas (4.38%).

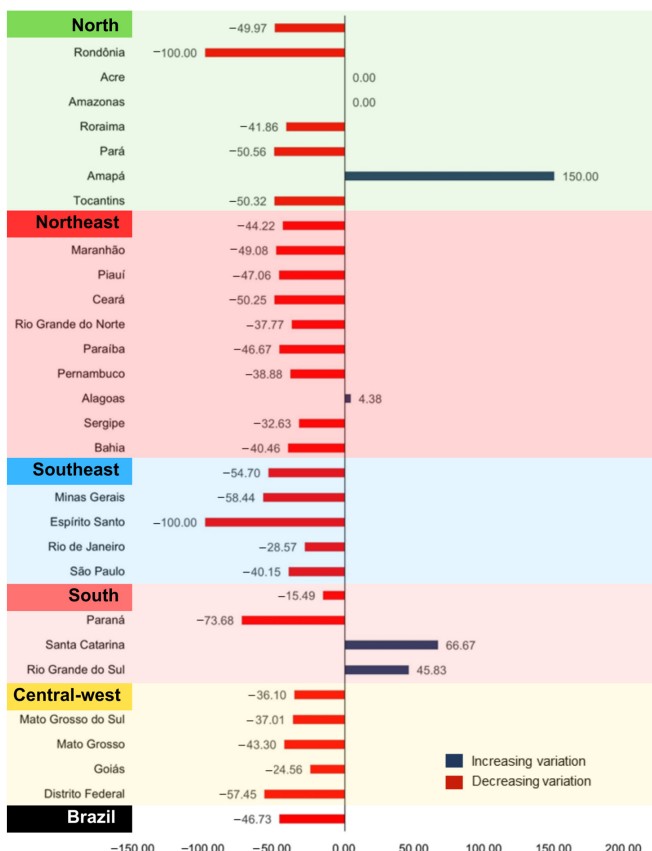

**Figure 2.** Description of the percentage change in new cases of VL by UF and Brazilian regions, 2020.

The predominant clinical-epidemiological characteristics of the study population were male gender (*n* = 1324; 68.53%), age group between 20 and 59 years old (*n* = 938; 50.88%), non-white skin color (*n* = 1608; 83.23%), and low educational level (*n* = 704; 36.44%). VL-HIV coinfection was present in 12.53% (*n* = 242) of cases. Regarding clinical evolution, 1259 cases were cured (65.17%) and 225 resulted in death, resulting in a lethality rate of 11.65%.

Figure 3 shows the number of new cases of COVID-19 and VL per month during the first year of COVID-19 pandemic. It is important to notice that since the beginning of pandemic, in March and April, there was an inverse relation between the number of new cases of COVID-19 and VL. Similarly, when there was a decrease in the notification of COVID-19 cases in August, a slight increase in VL new cases was observed over time (Figure 3).

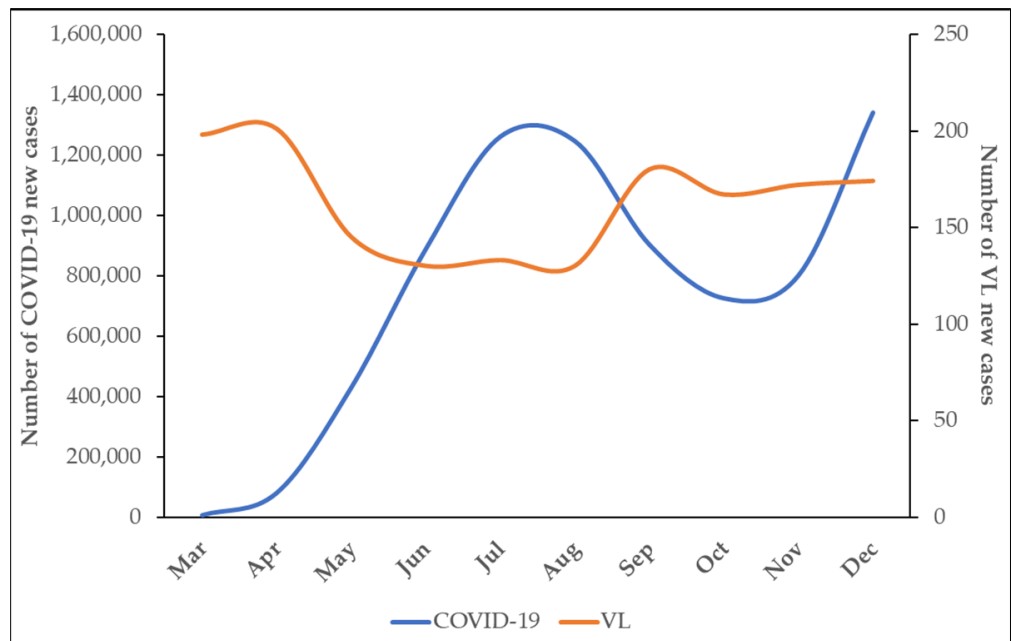

**Figure 3.** Monthly record of new cases of COVID-19 and VL, Brazil, 2020. Jan: January; Feb: February; Mar: March; Apr: April; May: May; Jun: June; Jul: July; Aug: August; Sep: September; Oct: October; Nov: November; Dec: December.

Regarding the monthly spatial distribution of the percentage variation in new cases of VL by state in 2020 (Figure 4), it was observed that most states had a decrease throughout the year, even before the spread of COVID-19 in the country (January to March). However, from May on, there was an increase in the number of states with a marked reduction in the percentage, especially in the Northeast and Southeast regions.

Although most states have shown a progressive reduction in the registration of new cases of VL, it is noticeable that in December, this trend underwent a modification. The North and Northeast regions reduced their negative percentual variation, while the South and Southeast regions showed a positive variation.

The choropleth maps of the spatial distribution of the incidence of VL at the municipal level are represented in Figure 5A,B. The average incidence of VL between the years 2015–2019 was widely distributed (Figure 5A), with greater concentration in the Northeast, Southeast, and Central-west regions. In Figure 5B, the average incidence of VL in the year 2020 can be observed.

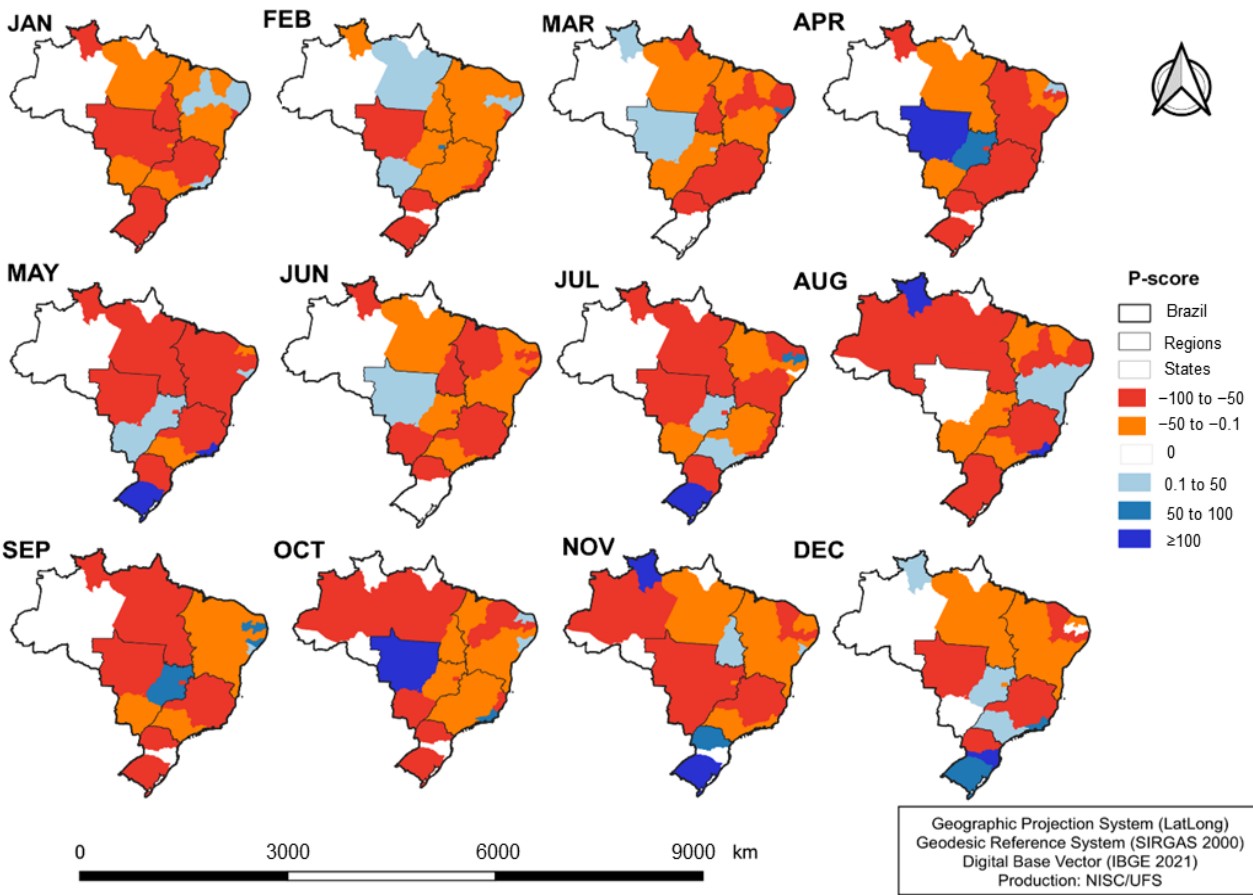

**Figure 4.** Monthly spatial distribution of percent change in new cases of VL by states in Brazil, 2020.
Jan: January; Feb: February; Mar: March; Apr: April; May: May; Jun: June; Jul: July; Aug: August;
Sep: September; Oct: October; Nov: November; Dec: December.

Considering the average of the last five years (2015–2019), just over one-third (36.3%) of
Brazilian municipalities reported any cases of VL. Of these, 47% (932) of the municipalities
are considered to have intense transmission ($\geq$4.4 cases/100,000 inhabitants). In 2020, only
786 municipalities (14.1%) reported new cases of VL. Compared to the average incidence
(2015–2019), there was a very significant reduction in new cases throughout the Brazilian
territory, as evidenced in Figure 5B. Furthermore, it is possible to observe a 42% reduction
in municipalities considered to have intense transmission.

A spatial autocorrelation analysis was obtained by calculating the univariate global
Moran's index, which revealed a statistically significant result, showing the existence
of spatial dependence of the occurrence of new cases of VL in the period of 2015–2019
($I = 0.491$; $p < 0.001$) and in 2020 ($I = 0.031$; $p = 0.009$). Figure 5C presents the Moran
map with clusters of municipalities in the period of 2015–2019, identified in the scatter
plot obtained by the univariate LISA. The high-risk areas in the period of 2015–2019
comprised 509 municipalities. On the other hand, Figure 5D represents the spatial clustering
analysis for the year 2020, in which the high-risk clusters comprised only 94 municipalities,
representing an 81.5% reduction in municipalities in this category.

A spatial–temporal scan analysis was performed in two steps in order to identify
clusters of high and low monthly incidence rates of VL throughout 2020. The high-risk and
low-risk spatial–temporal cluster were depicted in Figure 5E,F, whose characteristics are
described in Table 1.

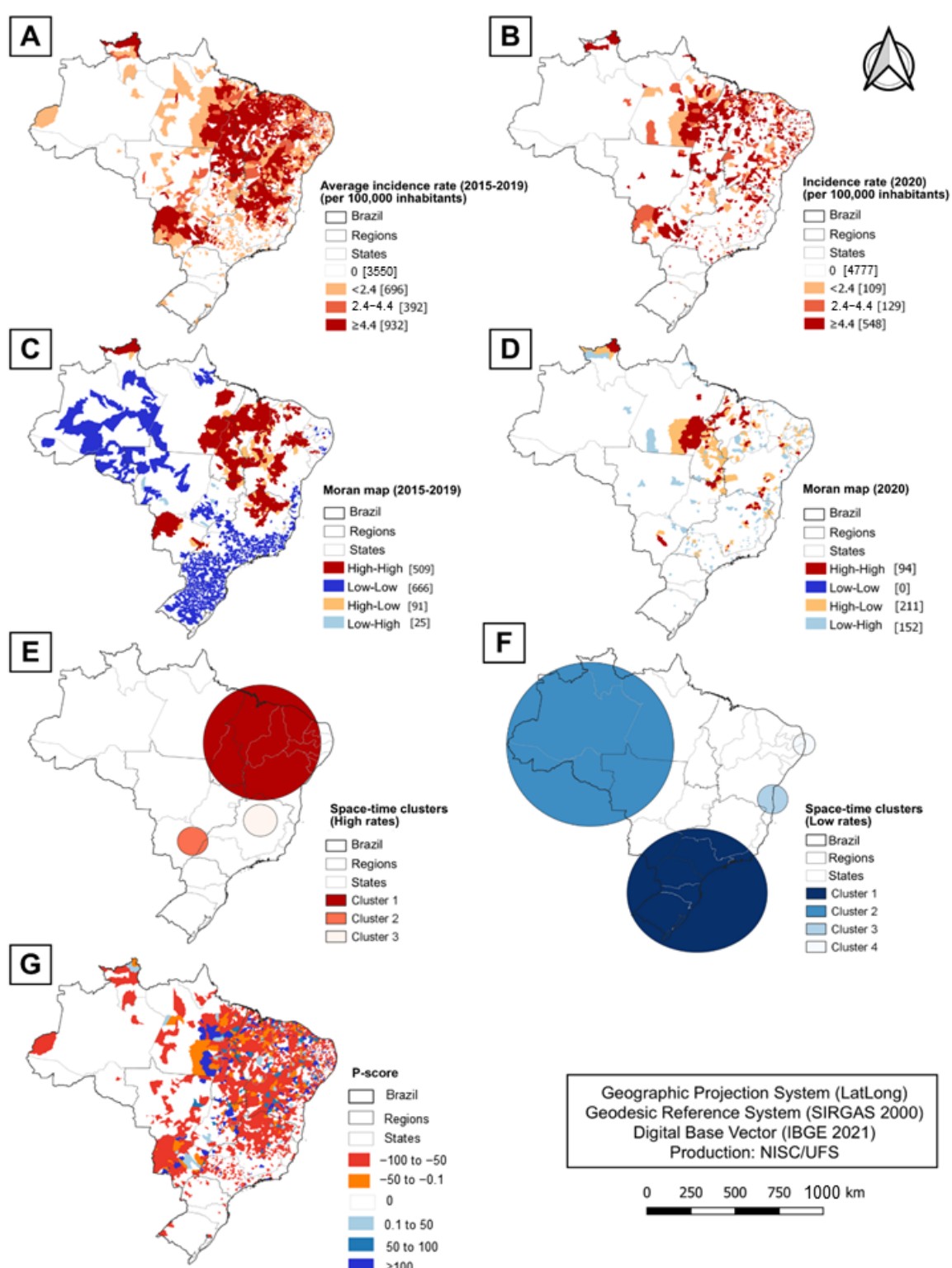

**Figure 5.** Spatial and spatiotemporal analysis of VL incidence in Brazil, 2015–2020. (**A**) Average crude incidence rate of VL (2015–2019); (**B**) crude incidence rate of VL (2020); (**C**) univariate LISA analysis of average crude incidence rate of VL (2015–2019); (**D**) univariate LISA analysis of crude incidence rate of VL (2020); (**E**) space–time scan analysis of monthly crude incidence rate of VL in 2020 (high rates); (**F**) space–time scan analysis of monthly crude incidence rate of VL in 2020 (low rates); (**G**) spatial distribution of P-score at the municipal level.

**Table 1.** Spatiotemporal clusters of VL incidence, Brazil, 2020.

| Cluster | Period | Number of Municipalities | Regions | New Cases | New Expected Cases | Monthly Incidence Rates [a] | RR | LLR |
|---|---|---|---|---|---|---|---|---|
| *Low rates* | | | | | | | | |
| 1 | March–August | 2349 | Central-west, Southeast, and South | 58 | 471 | 0.10 | 0.10 | 345.4 |
| 2 | January–June | 259 | North and Central-west | 1 | 50 | 0.02 | 0.02 | 46.2 |
| 3 | January–November | 116 | Northeast | 1 | 27 | 0.03 | 0.04 | 23.3 |
| 4 | March–August | 203 | Northeast | 9 | 44 | 0.20 | 0.20 | 21.4 |
| *High rates* | | | | | | | | |
| 1 | July–December | 1549 | Northeast, North, and Central-west | 673 | 185 | 3.30 | 5.06 | 457.4 |
| 2 | April–September | 50 | Central-west and Southeast | 48 | 8 | 5.40 | 6.05 | 45.9 |
| 3 | May–October | 170 | Southeast | 49 | 16 | 2.80 | 3.07 | 22.2 |

[a]: VL incidence rate (per 100,000 inhabitants) during the agglomeration period.

Regarding the percentage variations in the notification of new cases of VL in the first year of the COVID-19 pandemic (2020), it is possible to observe that 1598 municipalities (28.69%) presented a reduction in the number of notifications, while only 326 (5.85%) registered an increase, as shown in the thematic map in Figure 5G.

Notably, it is important to highlight the consistency of decreases in VL new case notifications in Brazil after 2020. Especially, in 2021, it was possible to notice that all Brazilian states registered a reduction in new cases. In 2022, only five states returned to increases in the notification of VL new cases (Figure S1). This finding is corroborated by the results of VL incidence mapping; it is evident that there was a progressive decrease in the number of municipalities classified as high-transmission areas with a parallel increase in the number of municipalities with no VL incidence during the COVID-19 pandemic (2020–2022) (Figure S2).

## 4. Discussion

Some studies have demonstrated the impact of the COVID-19 pandemic on the surveillance of infectious diseases worldwide, such as HIV, tuberculosis, and malaria in low- and middle-income countries [16], leprosy [17], tuberculosis [18], hepatitis C [19], and schistosomiasis [20] in Brazil. However, to the best of our knowledge, this is the first study that has investigated the impact of the COVID-19 pandemic on the notification of new cases of VL in Brazil.

The findings of this study demonstrate a significant reduction in the registration of new cases of VL in the country, especially in areas with higher incidence such as the Northeast, Southeast, and Central-west regions. The use of the P-score allowed for estimating the reduction in the notification of new cases of VL, and the integration of spatial and spatiotemporal analysis methods made it possible to identify the areas most affected by this reduction during the first year of the COVID-19 pandemic.

The clinical-epidemiological profile of the population in our study is similar to that described in previous studies. The prevalence in male individuals was described in the analysis of the epidemiological profile of 2155 cases of VL in the state of Pará between 2015 and 2019, as well as in the epidemiological analysis of VL in Picos, in the state of Piauí [21,22].

Assessments made by the WHO suggest that the pandemic has affected interventions in NTDs in three main areas: national-level implementation, management of the health product supply chain, including the delivery of medicines and diagnostics, and financial

resources. This shift in attention away from NTDs will certainly result in a negative impact on the healthcare system over the next few years [23,24]. Considering this, the results of this study corroborate those obtained in the study on the impact of the COVID-19 pandemic on the diagnosis of leprosy in Brazil, which found a 41.4% reduction in leprosy cases in Brazil in 2020, confirming the impact of the pandemic on NTDs [17]. The results of spatial and spatiotemporal analyses showed that VL is not randomly distributed in Brazil, as its distribution was found to be heterogeneous, with persistent cases in more vulnerable regions, but also with the formation of low-risk clusters in regions where incidence was much lower than expected.

In this context, it is possible to notice a convergence between the reduction in new cases of VL and the increase in the registration of new cases of COVID-19 throughout the epidemiological weeks of the year 2020. When analyzed retrospectively, it can be observed that the Southeast, Northeast, and North regions presented a significant growth in the number of cases and deaths [25], while they were the regions with the highest underreporting of new cases of VL.

It is important to mention that the underreporting of new cases of VL may be related to the measures implemented to mitigate the pandemic process, such as lockdowns, mobility restrictions, and social isolation. Such actions may have hindered the access of suspected cases to healthcare services. While in 2020, there was an increase in the average number of clinical consultations related to cutaneous leishmaniasis, clinical consultations for VL patients decreased considerably [26], which may have led to a reduction in the reporting of new cases.

Although there has been a reduction in the number of new cases of VL throughout the country, a high-risk cluster was observed to a greater extent in the Northeast region, followed by the North and Central-west regions. These data are a further reflection of the known susceptibility of vulnerable populations to NTDs, since all states in the Northeast region have poverty indicators above the national average [6,27].

Similar findings were observed in previous research, in which spatial and space–time clustering of VL cases was observed in the Northeastern region of Brazil with greater emphasis on its mid-north and hinterland sub-regions [6]. Additionally, a study on the spatial dynamics of tuberculosis in Brazil from March to December 2020 [18] revealed similar data. The percentage variation curve showed a progressive reduction in monthly notifications of new tuberculosis cases in all Brazilian regions, especially in the North, Northeast, and Central-west regions [18].

The impact of the COVID-19 pandemic on the surveillance of VL in Brazil, as highlighted in this ecological study, has several implications for global health.

### 4.1. Global Health Implications

Our study points to a substantial decrease in the notification of new VL cases in Brazil during the first year of the COVID-19 pandemic, which suggests that the Brazilian healthcare systems may have been overwhelmed by the demands of the pandemic, leading to reduced surveillance and reporting of other diseases. It is easy to imagine that a similar scenario may have occurred in other countries.

Furthermore, the negative percentage variations in VL registration observed in all Brazilian regions underline the importance of considering regional disparities in the response to global health crises like the COVID-19 pandemic, especially in large and diverse countries like Brazil.

Lastly, the study identifies spatial dependence of VL in Brazil, both before and during the pandemic. This highlights the interconnectedness of health issues within a country, which has implications for international health cooperation and information sharing to address cross-border health challenges.

*4.2. Limitations*

The results of this study should be interpreted considering certain limitations, primarily concerning the quality and timeliness of the data available in public records. The COVID-19 pandemic had a significant impact on healthcare infrastructure and data communication mechanisms, leading to delays in data entry. Consequently, it was not possible to include data from 2021 in our analysis, which could have provided additional insights into ongoing trends. On the other hand, despite the expressed limitations, it is important to highlight that we used an integrated methodological approach involving spatial and spatiotemporal analysis techniques and employed the P-score to estimate the percentage variations in VL notifications, which was the method used in previous studies to analyze the impact of the first year of the COVID-19 pandemic on notifications of leprosy [17], tuberculosis [18], hepatitis C [19], schistosomiasis [20], and meningitis [28]. Taken together, these aspects strengthened the methodological robustness of the study and can be replicated for the analysis of other public health issues.

**5. Conclusions**

The significant reduction in the notification of new cases of VL in Brazil during the first year of the COVID-19 pandemic is a concern for public health. Although the reduction may seem positive at first, it is important to emphasize that VL is a neglected tropical disease and potentially fatal when not treated promptly. Furthermore, the geographical overlap of diseases and the burden on epidemiological surveillance services due to the pandemic may have contributed to the decrease in VL notifications. The Southeast, North, and Northeast regions showed the highest negative percentage variations in the registration of new VL cases. Spatial analysis revealed the existence of spatial dependence of VL throughout the Brazilian territory in both periods, before and during the first year of the pandemic. However, there was a significant reduction in the number of municipalities composing the high-risk spatial cluster. This information is relevant for evaluating high-risk areas and can alert to the need to strengthen surveillance systems in the most vulnerable areas in order to develop more effective control and prevention strategies. It is important to highlight that, even during the pandemic, VL remains a concern for public health, and measures should be taken to ensure the appropriate diagnosis and treatment of the disease.

**Supplementary Materials:** The following supporting information can be downloaded at: https://www.mdpi.com/article/10.3390/idr16010009/s1, Figure S1: Spatial distribution of annual P-score of VL new cases notification. (A) 2015. (B) 2016. (C) 2017. (D) 2018. (E) 2019. (F) 2020. (G) 2021. (H) 2022, Figure S2: Spatial distribution of annual VL incidence. (A) 2015. (B) 2016. (C) 2017. (D) 2018. (E) 2019. (F) 2020. (G) 2021. (H) 2022.

**Author Contributions:** Conceptualization, C.J.N.R. and J.R.S.S.; methodology, C.J.N.R.; software, C.J.N.R.; validation, C.J.N.R., A.D.d.S. and S.V.M.A.L.; formal analysis, C.J.N.R. and S.V.M.A.L.; investigation, C.J.N.R., L.S.S. and J.R.S.S.; resources, Á.F.L.d.S., L.B.d.O. and I.A.C.M.; data curation, C.J.N.R.; writing—original draft preparation, J.R.S.S.; writing—review and editing, C.J.N.R., J.R.S.S., G.R.d.S.S., Á.F.L.d.S., L.B.d.O. and I.A.C.M.; visualization, Á.F.L.d.S. and I.A.C.M.; supervision, C.J.N.R.; project administration, J.R.S.S.; funding acquisition, Á.F.L.d.S. and I.A.C.M. All authors have read and agreed to the published version of the manuscript.

**Funding:** The study was supported by the Coordenacão de Aperfeiçoamento de Pessoal de Nível Superior—CAPES, Brazil; Process: 001.

**Institutional Review Board Statement:** Ethical review and approval were waived for this study due to this study using public-domain aggregate secondary data.

**Informed Consent Statement:** This study waived informed consent as it utilized anonymized and aggregated data from the public domain.

**Data Availability Statement:** Publicly available datasets were analyzed in this study. This data can be found here: https://datasus.saude.gov.br/ (accessed on 1 February 2024).

**Conflicts of Interest:** The authors declare no conflicts of interest.

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
