# Peer review of "Impact of the COVID-19 Pandemic Surveillance of Visceral Leishmaniasis in Brazil: An Ecological Study"

_2036-7449, doi:10.3390/idr16010009_

Round 1

Reviewer 1 Report (Previous Reviewer 1)

Comments and Suggestions for Authors

This study is looking at the impact of covid on visceral leishmaniasis, or its reporting. However, from the presented data it is not clear what this impact is and there are numerous flaws in their reasoning and the data. 

One major flaw is the authors do not deliniate whether the decrease in VL is due to a decrease reporting or an actual decrease in incidence. Futhermore, they waffle between these two options throughout the text making it very confusing for the reader. Sometimes they say it is decreased registration/reporting and sometimes they say it is a decreased incidence. So which is it? And how do they know? 

From the first sentence in the Conclusion section they lean towards a decreased reporting with the incidence being the same during covid as other years. No evidence for this is presented, though. Assuming underreporting is true, one would expect an increase in deaths or severe disease due to VL in 2021 and 2022. This data should be available. However, the authors refuse to include any data from these years. It is absolutely crucial to include the data from at least 2021 and preferably from both 2021-2022 as well. And perhaps even 2023 since we are almost in 2024. 

Along these same lines, I don't think it is appropriate to compare the average of 2015-2019 to a single year (2020). Are there any years during 2015-19 in which the incidence of VL is lower and comparable to the incidence in 2020? If so, this would drastically affect the interpretation and conclusions. Similarly, what are the P-scores in comparisons between the years 2015-2019? Do you see similar fluctuations in the P-scores in these post-covid years? Along with including data from post-covid years, a more detailed analysis of the pre-covid years is also necessary. A year-to-year fluctuation of leishmaniasis incidence is not unexpected and quite common. Many factors can influence the sandfly population and this often affects leishmaniasis incidence. 

The authors fail to discuss the possibility that the incidence did in fact decrease, and it was not just a decrease in reporting. For example, if leishmaniasis is associated with certain occupations or human activities that were shut down during the pandemic then there could have been an actual decrease in the incidence. 

Lines 357-362. The correlation between covid cases and VL cases is quite relevant to the theme of the paper and needs to be moved to the Results and elaborated upon. A graphic or table showing these results would be very helpful. 

How does Table 1 relate to the impact of covid -- the supposed theme of this paper? This table should be deleted since it is not related to the theme of the paper, or similar data from 2015-2019 should also be included so that a comparison can be made between pre- and post-covid. 

Numerous times throughout the manuscript and in figure legends the authors use pecentage change instead of P-score. The P-score is not a percentage change since the expected cases is a hypothetical number. Only P-score should be used. 

Figure 1 needs to be of higher resolution and a better legend would be: The geographical regions of Brazil. The legend should also define the state abbreviations. 

Lines 184-188 would be better at the end of the paragraph at line 174. 

Lines 238-241. Not clear what the authors are trying to say here. But I don't see any trends. It appears random. If there is a trend, which state(s) exhibit trends. For example, Roraima is red, orange, light blue, 3 reds in a row, orange, dark blue, red, white, dark blue, and light blue. What is the trend? Similarly, Bahia is either orange or red with one light blue. What is the trend? I didn't look at all the states, but I don't see any trends. 

lines 316-317, need a reference(s) for the 'few studies'.

lines 320-321, sentence should be deleted or moved to Introduction

The Discussion is rather long and contains a lot of material not directly relevant to the study. For example, lines 330-344 could be deleted.  

The section on Global Health Implications at line 386 seems rather redundant and repetitive. 

Author Response

Dear Editor and reviewers,

We hope this letter finds you well. We would like to express our sincere appreciation for the careful review and valuable feedback provided by the reviewers and yourself on our manuscript submitted to IDR. Your comments have been extremely helpful in improving the quality of our work.

We have taken the reviewers' comments into serious consideration and have made significant revisions to address the issues raised:

Question 1. One major flaw is the authors do not delineate whether the decrease in VL is due to a decrease reporting or an actual decrease in incidence. Furthermore, they waffle between these two options throughout the text making it very confusing for the reader. Sometimes they say it is decreased registration/reporting and sometimes they say it is a decreased incidence. So which is it? And how do they know?

Response 1: We acknowledge this observation. Nevertheless, there must be some incomprehension about the nature of our study design. In Brazil, the compulsory report is necessary for some diseases, such as visceral leishmaniasis. Thus, ecological studies use to be to consider the number of reports as a proxy of the disease’s incidence. Therefore, is plausible to use both expressions (“decrease of incidence” or “decrease of reporting”) since the decrease of reporting results in a decrease of incidence under a point of view of epidemiological analysis. Unfortunately, the ecological studies can not establish casual links between some exposures/independent variables and outcomes, such as incidence of a disease. Our study highlighted the possible reduction of VL incidence in the first year of COVID-19 pandemic. Probably, this decrease was not associated to a real reduction of the VL transmission in the territory, but due to a reduction of reports, since there was not any change of public health policy for the surveillance and control of the VL.

Question 2. From the first sentence in the Conclusion section, they lean towards a decreased reporting with the incidence being the same during covid as other years. No evidence for this is presented, though. Assuming underreporting is true, one would expect an increase in deaths or severe disease due to VL in 2021 and 2022. This data should be available. However, the authors refuse to include any data from these years. It is absolutely crucial to include the data from at least 2021 and preferably from both 2021-2022 as well. And perhaps even 2023 since we are almost in 2024.

Response 2: We appreciate your suggestions, but we must sustain our point of view. The scope of our study is focused on a specific topic: the decrease of reports of new cases/incidence of VL during the first year of COVID-19 pandemic. Including data of other years (e.g. 2021, 2022…) must be considered for further studies. Furthermore, the database used in this study was SINAN, which is not the system that encompasses data of mortality.

Question 3.  Along these same lines, I don't think it is appropriate to compare the average of 2015-2019 to a single year (2020). Are there any years during 2015-19 in which the incidence of VL is lower and comparable to the incidence in 2020? If so, this would drastically affect the interpretation and conclusions. Similarly, what are the P-scores in comparisons between the years 2015-2019? Do you see similar fluctuations in the P-scores in these post-covid years? Along with including data from post-covid years, a more detailed analysis of the pre-covid years is also necessary. A year-to-year fluctuation of leishmaniasis incidence is not unexpected and quite common. Many factors can influence the sandfly population, and this often affects leishmaniasis incidence.

Response 3: We thank the reviewer for this comment, but it is necessary to clarify some aspects of our statistical analysis in order to a better understanding of the nature of study. P-score calculation is well stablished in previous studies, such as da Paz et al. (2022), Souza et al. (2022), do Carmo et al. (2022), and Dantas et al. (2022), which were cited in our references. Although the reviewer believed that was not appropriate to compare the number of reported new cases in 2020 with the mean of 2015-2019, our choice was not random or arbitrary, but based on literature. Thus, there is no reason to carry out year-by-year comparison of p-scores of other years besides 2020. Additionally, we integrated different statistical methods of analysis, such as LISA and SatScan. Therefore, we believe that the doubts of the reviewer are answered with the results of other analysis.

Question 4. The authors fail to discuss the possibility that the incidence did in fact decrease, and it was not just a decrease in reporting. For example, if leishmaniasis is associated with certain occupations or human activities that were shut down during the pandemic then there could have been an actual decrease in the incidence.

Response 4: We believe that this question was partially answered with the Response 1. The decrease of VL crude incidence rates, the reduction of number of cities compounding high risk clusters, and negative p-scores in 2020 suggest that this reduction can be explained by the fact that the health system was focused on the coping of COVID-19 pandemic. We disagree that be plausible to discuss the possibility of a real decrease of VL incidence. VL is a vector-borne parasitic disease transmitted by phlebotomies that live indoor or in peridomicile areas due to anthropic modifications in its natural habitats. Thus, theoretically the lockdown or restriction of mobility could increase the exposure to the VL vector, leading to an increase in the VL incidence that would not be captured by the surveillance system.

Question 5. Lines 357-362. The correlation between covid cases and VL cases is quite relevant to the theme of the paper and needs to be moved to the Results and elaborated upon. A graphic or table showing these results would be very helpful.

Response 5: We agree with this suggestion and elaborated a graph depicting the relation between the number of new cases of COVID-19 and VL registered along 2020.

Question 6. How does Table 1 relate to the impact of covid -- the supposed theme of this paper? This table should be deleted since it is not related to the theme of the paper, or similar data from 2015-2019 should also be included so that a comparison can be made between pre- and post-covid.

Response 6: In order to address the previous question, we agree with this suggestion, and we deleted the Table 1, but we included a new figure that depicts the relation between the increase of new cases of COVID-19 and the decrease of VL new cases, since the patterns of clinical-epidemiological characteristics have not been changing along the time.

Question 7. Numerous times throughout the manuscript and in figure legends the authors use percentage change instead of P-score. The P-score is not a percentage change since the expected cases is a hypothetical number. Only P-score should be used.

Response 7: Actually, P-score is a percentage measure, since the formula includes a multiplication by a constant of 100. Therefore, the decimal P-score or the percentual change are interchangeable terms. The formula is described in the section method.

Question 8. Figure 1 needs to be of higher resolution and a better legend would be: The geographical regions of Brazil. The legend should also define the state abbreviations.

Response 8: We improved the Figure 1 that depicts the study area and added the legend describing the Brazilian states abbreviations.

Question 9. Lines 184-188 would be better at the end of the paragraph at line 174.

Response 9: We reordered the sentence as suggested.

Question 10. Lines 238-241. Not clear what the authors are trying to say here. But I don't see any trends. It appears random. If there is a trend, which state(s) exhibit trends. For example, Roraima is red, orange, light blue, 3 reds in a row, orange, dark blue, red, white, dark blue, and light blue. What is the trend? Similarly, Bahia is either orange or red with one light blue. What is the trend? I didn't look at all the states, but I don't see any trends.

Response 10: We believe that the Figure 3 helps the comprehension of Figure 4. It is notably visible the reduction of number of states that showed and negative variation of registration of VL new cases, especially in December. Therefore, the findings must be interpreted together.

Question 11. lines 316-317, need a reference(s) for the 'few studies'.

Response 11: We did not find studies that have investigated the impact of COVID-19 pandemic in notification of VL new cases in Brazil. So, we rewrote the sentence as following: “However, to the best of our knowledge this is the first study that have investigated the impact of the COVID-19 pandemic on the notification of new cases of VL in Brazil.”

Question 12. lines 320-321, sentence should be deleted or moved to Introduction

Response 12: We excluded the sentence as suggested.

Question 13. The Discussion is rather long and contains a lot of material not directly relevant to the study. For example, lines 330-344 could be deleted. 

Response 13: We remove the lines from the discussion as suggested by the reviewer.

Question 14. The section on Global Health Implications at line 386 seems rather redundant and repetitive.

Response 14: Since we have substantially reduced the discussion, as suggested by the reviewer, we believe that the notes outlined in this section provide a comprehensive and clear overview of our findings, as it highlights the fragility of the health system, especially epidemiological surveillance and control of infectious diseases to deal with the burden imposed by the geographic overlap of different diseases, especially in an extraordinary pandemic situation.

Reviewer 2 Report (Previous Reviewer 2)

Comments and Suggestions for Authors

The authors of the article addressed most of the comments of the reviewers and thus the revised article is more comprehensible and provides a clear view of the discussed subject. As far as the rest of the comments that could not be addressed, the authors clarified in the text the limitations and the problems of such a study in the discussion section. I believe that the revised manuscript provides an interesting and well-provided article for the impact and the significance of the COVID-19 pandemic in the more and more decreased surveillance of a neglected disease such as the visceral leishmaniasis in an endemic country.

Author Response

Thank you very much for your feedback!

Round 2

Reviewer 1 Report (Previous Reviewer 1)

Comments and Suggestions for Authors

The reasons for not including data from 2021-23 and not showing the individual years from 2015-2019 are not compelling. Such data would greatly enhance the value of this publication. Therefore, my recommendation remains to reject the paper. 

Furthermore, this refusal to do relatively little work which potentially increases the significance of the study raises suspicions. Why not do these analyses? 

Author Response

Dear Reviewer,

We sincerely appreciate the opportunity to review our manuscript and the valuable suggestions provided by the reviewers. In response to the comments from Reviewer 1 and the Associate Editor, we have made significant changes to enhance the quality and scope of our study.

Addressing the concern raised by Reviewer 1 regarding the exclusion of data from 2021-23 and the non-presentation of individual years from 2015-2019, we have now included, as supplementary material, the databases corresponding to these periods. However, it is important to note that the LV notification data for the year 2023 is currently unavailable in the official government database, making it inaccessible at this moment due to the time-consuming procedures of case investigation and confirmation, usually consolidated only in the subsequent year.

Additionally, in response to the suggestion from the Associate Editor, we have included two illustrative figures representing the spatial distribution of the annual p-score and the incidence of LV from 2015 to 2022. This analysis strengthens our initial hypothesis of a reduction in the notification of new cases, without necessarily indicating changes in public policy for disease surveillance and control. We emphasize that the redirection of efforts toward controlling the COVID-19 pandemic may have played a significant role in this scenario.

While we understand the importance of extending the analysis period, our decision to focus on the first year of the pandemic is grounded in the pursuit of a clear and concise analysis. We acknowledge the value of extending the temporal range, but we believe this suggestion aligns more appropriately with subsequent investigations, as mentioned in our previous response.

We sincerely hope that these changes address the concerns raised by the reviewers and the Associate Editor, and that our study will be deemed suitable for publication in the IDR journal. We believe our contributions can enhance the understanding of the impacts of the pandemic on the control of neglected tropical diseases, such as LV.

Thank you once again for your consideration, and we eagerly await the opportunity to publish our work in your esteemed journal.

Best regards,

This manuscript is a resubmission of an earlier submission. The following is a list of the peer review reports and author responses from that submission.

Round 1

Reviewer 1 Report

Comments and Suggestions for Authors

This is a paper looking at the reporting of visceral leishmaniasis during 2020 and implying an impact of covid19. 

There are major flaws in this study and it is incomplete. 

One flaw is the use of only data from 2020. Data from 2021 and 2022 would be very informative. Since they used data from 2015-2019 as pre-covid, it would make sense to also use 5 years post-covid (ie, 2020-2024). But at a minimum the data from 2021-2022 should be included. 

The authors assumed the decrease was solely due to reporting. Could it be that there was less transmission due to the economic shutdown during the pandemic? This is never discussed. And, if it is solely reporting, one would expect more deaths from VL. What is the mortality due to VL during 2015-2022? If cases are not being diagnosed, one would expect more deaths. 

There is an issue with formatting, Figure 2 is on top of Table 1. 

Reviewer 2 Report

Comments and Suggestions for Authors

In this article, the authors tried to address the problem of underregistered cases of the neglected disease of visceral leishmaniasis in Brazil in 2020 due to the COVID-19 pandemic. Taking into consideration the 5,570 Brazilian municipalities'  new VL registrations in 2020 and compared them with the corresponding incidences of the yeras 2015-2019, the authors performed analyses of p-score, global and local univariate Moran's Indices and retrospective space-time scan statistics. Based on those analyses, the authors pointed out that 46.73% less new cases of VL in total in Brazil during 2020 were reported than expected.  By highlighting this public health concern, the authors provide information about the evaluating high-risk areas and populations in Brazil which can be  alerted for strengthening surveillance  systems and  developping more effective control and prevention strategies. I believe that this article will be beneficial for the readers of infectious disease rfeports but before publishing, it should be recontructed in order for:

a) Figure 2 not to overlap with Table 1 

b) Figure 2 to be more clear about the subcategories of the regions

c) Data of Table 1 to be more extensively elaborated in the result and discusssion sections.